# Re-weighting Tokens: A Simple and Effective Active Learning Strategy for Named Entity Recognition

**Haocheng Luo, Wei Tan, Ngoc Dang Nguyen** and **Lan Du**[*]
Department of Data Science and AI, Monash University
{Haocheng.Luo, Wei.Tan2, dan.nguyen2, Lan.Du}@monash.edu

## Abstract

Active learning, a widely adopted technique for enhancing machine learning models in text and image classification tasks with limited annotation resources, has received relatively little attention in the domain of Named Entity Recognition (NER). The challenge of data imbalance in NER has hindered the effectiveness of active learning, as sequence labellers lack sufficient learning signals. To address these challenges, this paper presents a novel reweighting-based active learning strategy that assigns dynamic smoothed weights to individual tokens. This adaptable strategy is compatible with various token-level acquisition functions and contributes to the development of robust active learners. Experimental results on multiple corpora demonstrate the substantial performance improvement achieved by incorporating our re-weighting strategy into existing acquisition functions, validating its practical efficacy.

## 1 Introduction

The effectiveness of deep neural networks has been well-established on diverse sequence tagging tasks, including named entity recognition (NER) (Lample et al., 2016). However, the evolving complexity of these models demands substantial labeled data, making annotation a resource-intensive endeavor. Active learning (AL) addresses this by astutely focusing on information-rich sequences, substantially reducing data that needs expert annotation. Such a strategy proves invaluable in NER, as it brings down costs and accelerates learning by honing in on complex or dynamic entities. AL's iterative process smartly picks subsets from an unlabeled pool, leveraging expert-annotated data to refine the model. Modern AL approaches primarily pivot on model uncertainty (Houlsby et al., 2011; Gal et al., 2017; Shen et al., 2017; Kirsch et al., 2019; Zhao et al., 2021) or data diversity (Sener and Savarese,

_______________
\* Corresponding Author

2017; Shen et al., 2017; Tan et al., 2021; Hacohen et al., 2022).

The primary challenge in applying active learning to sequence tagging lies in addressing data imbalance at the entity level. Traditional methods of active sequence tagging, such as those referenced by (Shen et al., 2017; Zhang et al., 2020; Shelmanov et al., 2021) , typically generate scores for sentences by summing or averaging the tokens within them, thereby treating each token equally. Radmard et al. (2021) attempted to address this by segmenting sentences for token-level selections, However this led to loss of context and semantic meaning, impairing human understanding. To effectively tackle this data imbalance issue, it is imperative to focus on the token level within the sentences. By implementing a strategy that redistributes sentence selection through assigning weights to the tags of the tokens, a more balanced and efficient handling of the data can be achieved.

In supervised learning (SL) and semi-supervised learning (SSL), re-weighting is a widely researched method to tackle data imbalance. However, they have limitations to be directly applied to active learning. For example, Wei et al. (2021); Cao et al. (2019) assume the labeled data and the unlabeled data have the same label distribution, while we usually have no prior information about the unlabeled data in active learning; Kim et al. (2020) use the confusion matrix on the labeled data to approximate the label distribution of the unlabeled data. In active learning, the labeled data pool can be quite small, so the approximation can cause strong biases; Li et al. (2018) suggest a gradient-based method to compute weights, while computing gradients significantly increases the computational cost for active learning.

In this paper, we focus on the feasibility of adapting reweighting-based methods from SL, SSL to active sequence tagging, which has not been developed in existing works. Our contributions can be

summarized as follows:

- We propose a flexible reweighting-based method for active sequence tagging to tackle the data imbalance issue. It is easy to implement, and generic to any token-level acquisition function.
- We conduct extensive experiments to show our method can improve the performance for each baseline, across a large range of datasets and query sizes. We also show that it can indeed mitigate the class imbalance issue, which explains its success.

## 2 Related works

**Active learning for NER.** Active learning for NER has seen a variety of methodologies being developed to address its unique challenges (Settles and Craven, 2008; Marcheggiani and Artieres, 2014; Shelmanov et al., 2021). The overarching goal is to reduce the budget for labeling sequence data by selectively querying informative samples. The information content of a sample is typically measured using an acquisition function, which quantifies the potential usefulness of unlabeled data.

Uncertainty-based acquisition functions have gained popularity due to their successful application and their compatibility with neural networks, which can generate a probability distribution using the softmax function over the model's output. Various acquisition functions have been proposed to represent uncertainty. Least Confidence (LC) (Li and Sethi, 2006), for example, selects the sentence with the lowest model prediction confidence, while Sequence Entropy (SE) (Settles and Craven, 2008) employs the entropy of the output instead of the probability. Alternatively, Margin (Scheffer et al., 2001) defines uncertainty as the difference between the highest and the second-highest probability.

These traditional approaches, however, have their limitations. Shen et al. (2017) pointed out that summing over all tokens, as done in previous methods, can introduce bias towards longer sentences. To overcome this, they proposed Maximum Normalized Log-Probability (MNLP), a normalized version that mitigates this issue. In a more recent development, Lowest Token Probability (LTP) (Liu et al., 2022) selects the sentence containing the token with the lowest probability. Additionally, Bayesian Active Learning by Disagreement (BALD) (Gal et al., 2017) uses Monte-Carlo Dropout to approximate the Bayesian posterior, which makes it feasible to approximate the mutual information (Houlsby et al., 2011) between outputs and model parameters.

It is worth noting that we exclude subsequence-based active learning methods (Wanvarie et al., 2011; Radmard et al., 2021; Liu et al., 2023) from our comparison. In their setting, the query is a subsequence instead of the whole sentence. As they admitted, it may be not practical in the real world because subsequences can be meaningless. Human experts usually need the context to label tokens.

**Re-weighting methods for supervised learning and semi-supervised learning.** Many real-world datasets exhibit a "long-tailed" label distribution (Van Horn et al., 2018; Liu et al., 2020; Chu et al., 2020). Imbalanced supervised learning has been widely researched. Re-weighting the loss (Khan et al., 2017; Aurelio et al., 2019; Tan et al., 2020) is one of the popular methods on this topic. In recent semi-supervised learning methods, re-weighting is gradually gaining attention. Wei et al. (2021) utilize the unlabeled data with pseudo labels according to an estimated class distribution, Kim et al. (2020) develop a framework to refine class distribution, Lai et al. (2022) use the effective number defined by Cui et al. (2019) to produce adaptive weights.

## 3 Methodology

### 3.1 Preliminaries

Consider a multi-class token classification problem with $C$ classes. An input is a sequence of $T$ tokens, denoted by $\mathbf{x} = \{x^1, ...x^T\}$, and its corresponding label is denoted by $\mathbf{y} = \{y^1, ...y^T\}$. The labeled data contains $m$ samples, denoted by $\mathcal{L} = \{\mathbf{x}_i, \mathbf{y}_i\}_{i=1}^m$, and the unlabeled data contains $n$ samples, denoted by $\mathcal{U} = \{\mathbf{x}_j\}_{j=1}^n$, with $m << n$.

In each active learning iteration, we select $B$ data points to form a query $\mathcal{Q}$, according to some acquisition function $q$:

$$\mathbf{x}^* = \arg\max_{\mathbf{x} \in \mathcal{U}} q(\mathbf{x})$$

### 3.2 Re-weighting Active learning for named entity recognition

Inspired by reweighting-based methods for supervised learning and semi-supervised learning, we propose a new re-weighting strategy for active sequence tagging, which assigns a smoothed weight for each class, which is inversely proportional to

the class frequency in the labeled data pool:

$$w_k = \frac{1}{m_k + \beta m}, \; for \; k = 1, 2, ....C$$

where $\beta$ is a hyperparameter controlling the degree of smoothing, $m_k = \sum_{(x,y)\in\mathcal{L}} \mathbb{1}_{y=k}$ is the sample size of class $k$ in the labeled data pool, $m$ represents the size of the entire labeled data pool. When $\beta = 0$, weights are strictly the inverse of the class frequencies; when $\beta = \infty$, weights follow a uniform distribution, i.e. no re-weighting is activated. A reasonable value for $\beta$ plays a significant role to combat sampling bias at the early stage of active learning. Based on this re-weighting function,

---

**Algorithm 1** Re-weighting active learning for NER

---

**Require:** $Neural\; network\; f(x;\theta),\; labeled\; pool\; \mathcal{L},$
  $unlabeled\; pool\; \mathcal{U},\; size\; of\; queries\; B,$
  $a\; base\; token-level\; acquisition\; function\; q.$
  computing weights using the labeled data

$$w_k = \frac{1}{m_k + \beta m}, \; for \; k = 1, 2, ....C$$

Initialize query set: $\mathcal{Q} = \emptyset$
**for** $\mathbf{x} \in \mathcal{U}$ **do**
  $\hat{y}^t = \arg\max_c f(y_c|x^t; \theta)$
  $q(\mathbf{x}) = \sum_t w_{\hat{y}^t} q(x^t)$
**end for**
**for** $b = 1, 2..., B$ **do**
  $\mathbf{x}^* = \arg\min_{\mathbf{x}\in\mathcal{U}}(q(\mathbf{x}))$
  $\mathcal{Q} = \mathcal{Q} \cup \mathbf{x}^*$
  $\mathcal{U} = \mathcal{U} \setminus \mathbf{x}^*$
**end for**
**return** $\mathcal{Q}$

---

we generate a sentence-level acquisition score by computing the weighted sum over all tokens. The algorithm is shown in Algorithm 1. It should be noticed that we do not have true labels of unlabeled data in active learning, therefore we use the pseudo label $\hat{y}$ instead. This pseudo label trick is widely used and has been verified as effective in many active learning works (Shen et al., 2017; Ash et al., 2019; Wang et al., 2022; Mohamadi et al., 2022).

Our method has the following advantages: **Independent on the label distribution of the unlabeled data.** Estimating the label distribution of the unlabeled data can be tricky, especially at the early stage of active learning or the unlabeled data does not have the same distribution as the labeled data. Compared to reweighting-based methods for SSL (Wei et al., 2021; Lai et al., 2022),

we do not need to make assumptions on the unlabeled data. **Computationally lightweight.** The time complexity of computing weights is $\mathcal{O}(m)$. Note that $m << n$ in active learning, so it is more efficient compared to the effective number-based reweighting-based method with time complexity $\mathcal{O}(n)$ (Lai et al., 2022). **Generic to any base acquisition function.** Since our method is essentially a weighted sum strategy, it can be combined with any base acquisition function for active learning mentioned in section 4.

## 4 Experimental Setups

**Datasets:** In this section, we evaluate the efficacy of re-weighted active learning on three widely used NER datasets, which are listed as follows, the statistical data of each dataset is provided in the Appendix B.
- **Conll2003** (Sang and De Meulder, 2003) is a corpus for NER tasks, with four entity types in BIO format (LOC, MISC, PER, ORG), resulting $C = 9$ classes.
- **WikiAnn (English)** (Pan et al., 2017) is a corpus extracted from Wikipedia data, with three entity types in BIO format (LOC, PER, ORG), resulting $C = 7$ classes.
- **BC5CDR** (Wei et al., 2016) is a biomedical dataset, with two entity types in BIO format (Disease, Chemical), resulting $C = 5$ classes.

**NER model:** We finetuned BERT-base-cased model (Devlin et al., 2018) with an initialized linear classification layer to perform sequence tagging. All experiments share the same settings: we used Adam (Kingma and Ba, 2014) with the constant learning rate at $5e - 5$ and the weight decay at $5e - 5$ as the optimizer; the number of training iterations is 30. All experiments were done with an Nvidia A40 GPU.

**Active learning settings:** The initial size of the labeled data pool is 30, the number of active learning iterations is 10, the query size varies from $\{15, 30, 50\}$. In each active learning iteration, we selected a batch of unlabeled data, queried their labels, added them to the labeled data pool, and then re-trained the model. We evaluated the performance of algorithms by their mean F1-scores with $95\%$ confidence interval for five trials on the test dataset.

**Baselines:** We considered five baselines: (1) randomly querying, (2) Least Confidence (LC) (Li and Sethi, 2006), (3) Maximum Normalized Log-

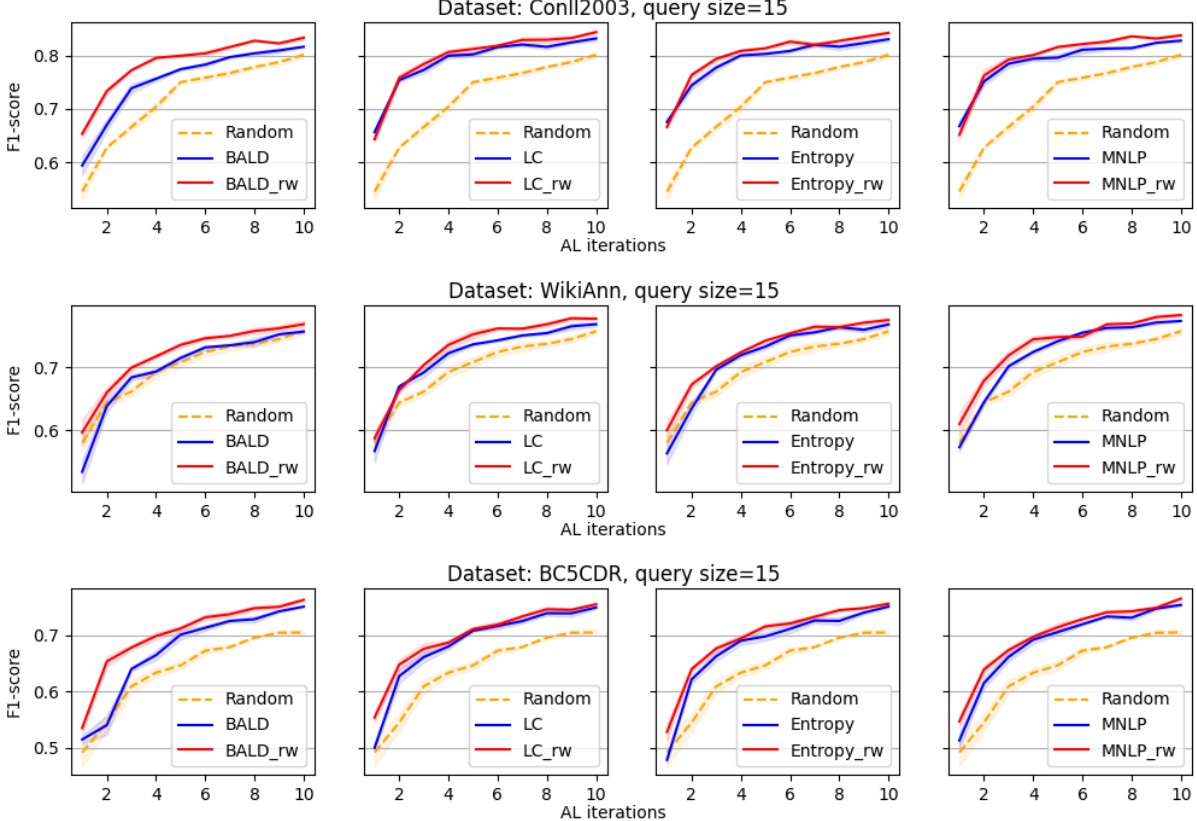

Figure 1: Pairwise comparison between with and without re-weighting for each baseline with query size 15. 'rw' is an abbreviation for re-weighting. Orange dashed lines indicate randomly querying, blue solid lines represent the original baselines, red solid lines represent the reweighting-based versions. The shaded area indicates the 95% confidence interval.

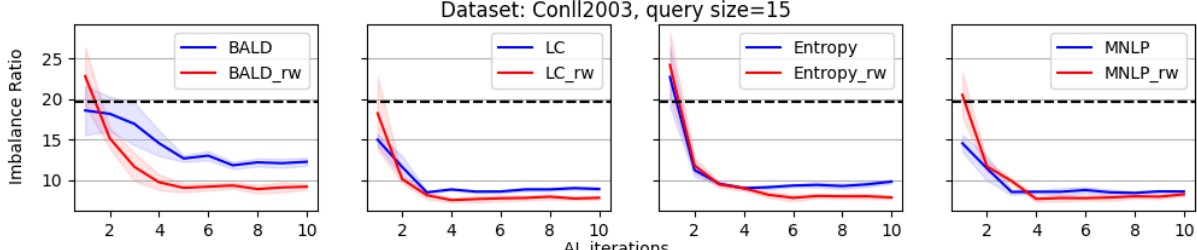

Figure 2: The variation of the imbalance ratio during active learning process on Conll2003 with query size 15. Black dashed lines indicate the overall imbalance ratio of the entire dataset, blue solid lines represent original baselines, red solid lines represent the reweighting-based versions. The shaded area indicates the 95% confidence interval.

Probability (MNLP) (Shen et al., 2017), (4) Sequence Entropy (SE) (Settles and Craven, 2008), (5) Bayesian Active Learning by Disagreement (BALD) (Gal et al., 2017). The details of the baselines are mentioned in the Appendix A.

**Hyperparameter setting:** As our reweighting-based methods have a hyperparameter $\beta$ to be tuned, we first conduct a grid search over $\{0.01, 0.1, 0.2, 0.5, 1\}$ in the re-weighting LC case on each dataset, and fix the optimal value $\beta = 0.1$

for all experiments. Please see Appendix C for the performances of all hyperparameters.

## 5   Experimental Results & Discussions

### 5.1   Main results

We evaluate the efficacy of our reweighting-based method by comparing the performance of active sequence tagging with and without re-weighting. Figure 1 shows the main results on three datasets with query size 15. We delay the results of other

query sizes to Appendix C. It is clear that each re-weighting method consistently outperforms the original method across different datasets and query sizes.

## 5.2 Analysis of the performance gain

As we discussed, the label imbalance issue occurs commonly in NER datasets, and it potentially damages the performance. We argue that the performance gain of the reweighting-based method is because it can effectively decrease the imbalance ratio. To evaluate this, we define an imbalance ratio as

$$\gamma = \frac{1}{C} \sum_c \frac{N_c}{N_{min}},$$

where $N_c$ is the sample size of class $c$ and $N_{min}$ is the sample size of the class with the minimum number of samples in the labeled dataset. For a balanced label distribution, the imbalance ratio is close to 1. A larger imbalance ratio indicates a higher class imbalance.

As shown in Figure 2, we plot the variation of the imbalance ratio on Conll2003 during the active learning process. Results on other datasets are put in Appendix C due to space limitations. In general, models have a lower confidence on samples from minority classes. As a result, uncertainty-based acquisition functions bias towards selecting samples from minority classes implicitly. This explains why they can achieve better performance than randomly querying. In contrast, reweighting-based methods further bias towards minority classes explicitly. It leads to more balanced querying, which can cause better performance.

## 5.3 Ablation study

**Effect of smoothing.** The first term in the denominator of our re-weighting function $\beta m$, which controls the degree of smoothing, plays a significant role at the early stage of active learning. Not using the smoothed version may encounter sampling bias issues. To verify these views, we report f1-scores of the smoothed version and the non-smoothed version in the first three AL iterations in table 1, across five independent runs. For each experiment, we set LC as the base acquisition function. It can be observed that non-smoothed version has lower performance and higher variance at first. This is what we expect to see in our motivation.

| Iteration# | 1 | 2 | 3 |
|---|---|---|---|
| Smoothed ($\beta = 0.1$) | 61.2±0.2 | 76.5±0.9 | 81.0±0.3 |
| Non-smoothed | 60.3±1.0 | 68.9±1.8 | 78.2±1.3 |

Table 1: Comparison with non-smoothed re-weighting function. Mean F1-scores and their $95\%$ confidence intervals in the first three AL iterations on Conll2003 with query size 15 are reported.

## 6 Conclusion

In this work, we pioneered the use of re-weighting in active learning for NER to tackle the label imbalance issue. Our method uses the class sizes in the labeled data pool to generate a weight for each token. We empirically demonstrate that our reweighting-based method can consistently improve performance for each baseline, across a large range of NER datasets and query sizes.

## 7 Limitations

We acknowledge that the static nature of the hyperparameter $\beta$ is a limitation of our work, particularly when applying the algorithm to new or diverse datasets. Dynamically updating $\beta$ in line with Active Learning iterations can be a promising avenue for future research.

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

# A  Baselines

For completeness, we list several commonly used acquisition functions below.

Least Confidence (LC) (Li and Sethi, 2006) uses the probability of the model's prediction to measure the uncertainty:

$$q_{LC}(\mathbf{x}) = \sum_t q_{LC}(x^t) \tag{1}$$

$$= \sum_t 1 - p(\hat{y}^t | x^t, \theta) \tag{2}$$

Sequence Entropy (SE) (Settles and Craven, 2008) utilizes the probability distribution instead of the probability to measure the uncertainty:

$$q_{SE}(\mathbf{x}) = \sum_t q_{SE}(x^t) \tag{3}$$

$$= \sum_t \sum_c p(y_c^t | x^t, \theta) log p(y_c^t | x^t, \theta) \tag{4}$$

Bayesian Active Learning by Disagreement (BALD) (Gal et al., 2017) utilizes MC-dropout samples to approximate the mutual information between outputs and model parameters:

$$q_{BALD}(\mathbf{x}) = \sum_t q_{BALD}(x^t) \tag{5}$$

$$= \mathbb{I}[y, \theta | x, \mathcal{L}] \tag{6}$$

$$= \mathbb{H}[y | x, \mathcal{L}] - \mathbb{E}_{p(\theta | \mathcal{L})}[\mathbb{H}[y | x, \theta]] \tag{7}$$

$$= -\sum_{t,c} (\frac{1}{M} \sum_m p_{c,t}^m) log(\frac{1}{M} \sum_m p_{c,t}^m) \tag{8}$$

$$+ \frac{1}{M} \sum_t \sum_{c,m} p_{c,t}^m log p_{c,t}^m \tag{9}$$

where $\hat{y}$ is the prediction of the model, $M$ denotes MC-dropout samples. Different from the acquisition functions mentioned above, which sum over all tokens, Maximum Normalized Log-Probability (MNLP) (Shen et al., 2017) normalized by the length of the sequence:

$$q_{MNLP}(\mathbf{x}) = \frac{1}{T} \sum_t log p(\hat{y}^t | x^t, \theta)$$

# B  Dataset statistics

See table 3 for the statistical data for each dataset.

# C  Additional results

In this section, we present all the remaining experimental results that could not be showcased in the main paper due to the space limitation.

| Dataset | Sentences | Tokens | Average length | Class proportion (B/I/O) | Imbalance ratio |
|---------|-----------|--------|----------------|--------------------------|-----------------|
| Conll2003 | 14041 | 203,621 | 14.5 | 11.5%/5.2%/83.3 | 19.7 |
| WikiAnn | 20000 | 160394 | 8.0 | 17.4%/31.9%/50.7% | 2.35 |
| BC5CDR | 5228 | 109322 | 20.6 | 8.6%/2.9%/88.5% | 38.4 |

Table 2: Statistics for the entire training dataset of the three datasets used in our experiments.

| $\beta$ | 0.01 | 0.1 | 0.2 | 0.5 | 1 |
|---------|------|-----|-----|-----|---|
| Conll2003 | 84.28 | **84.54** | 84.45 | 84.33 | 83.37 |
| WikiAnn | 78.18 | **78.53** | 77.32 | 77.60 | 76.15 |
| BC5CDR | 75.50 | **75.98** | 75.90 | 75.63 | 75.51 |

Table 3: Final F1-scores of re-weighting LC with different hyperparameter $\beta$ on each dataset.

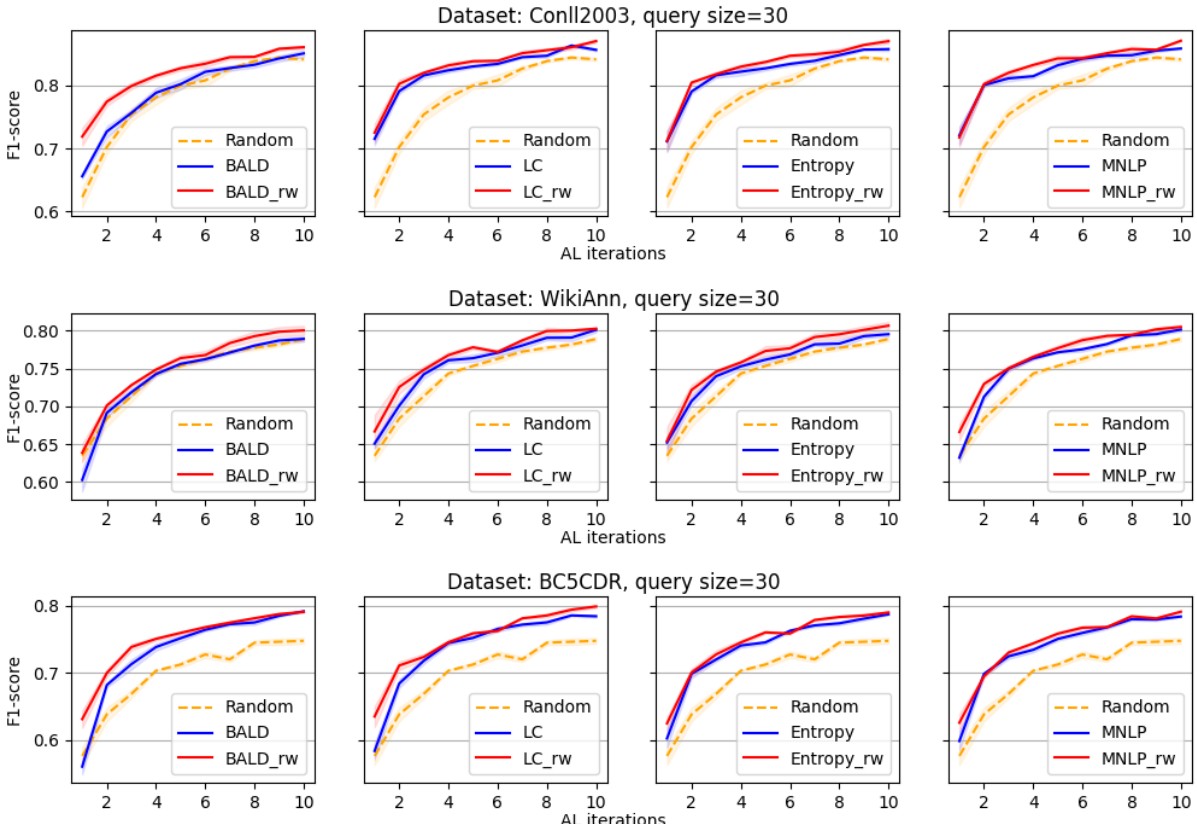

Figure 3: Pairwise comparison between with and without re-weighting for each baseline with query size 30.

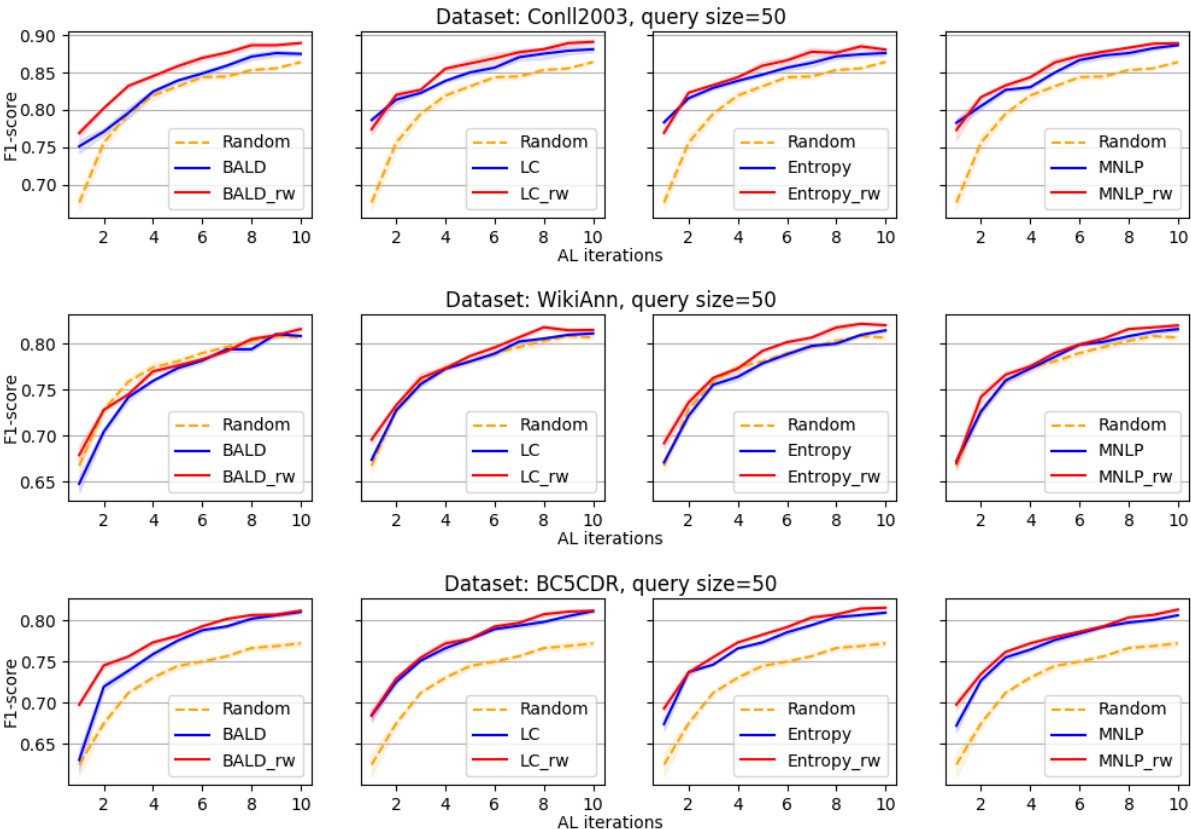

Figure 4: Pairwise comparison between with and without re-weighting for each baseline with query size 50.

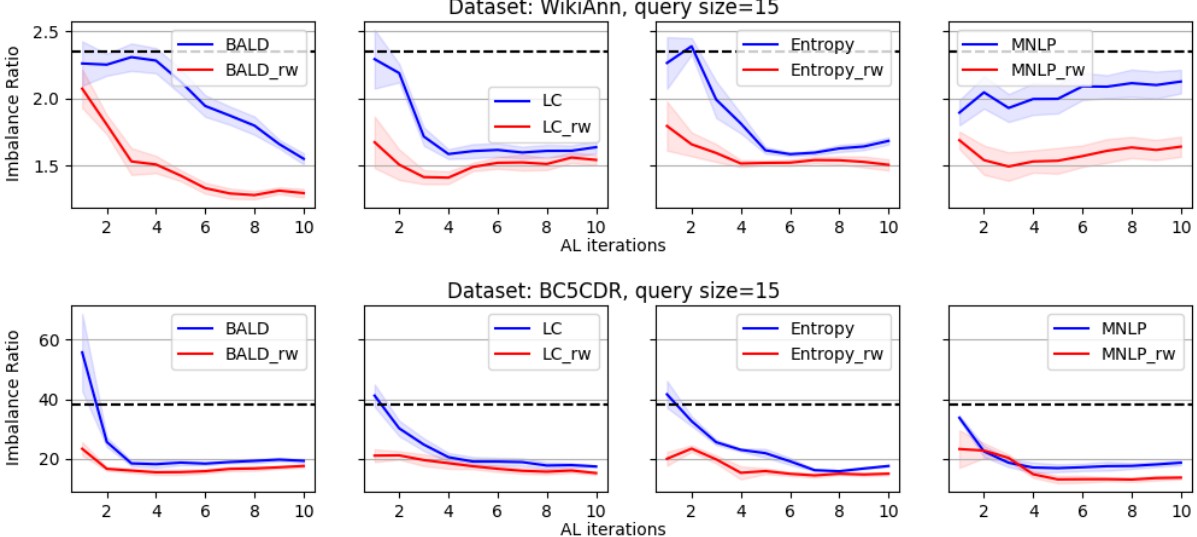

Figure 5: The variation of the imbalance ratio during the active learning process on WikiAnn and BC5CDR with query size 15.

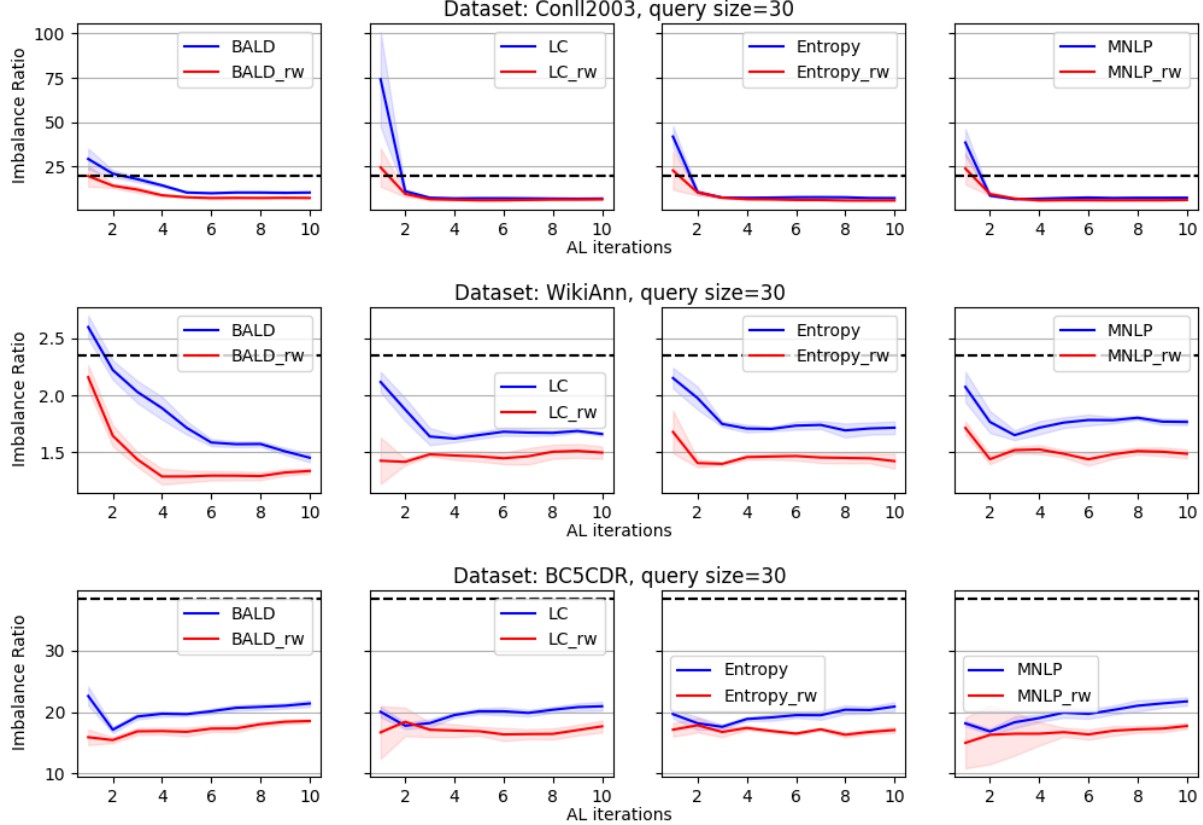

Figure 6: The variation of the imbalance ratio during the active learning process on three datasets with query size 30

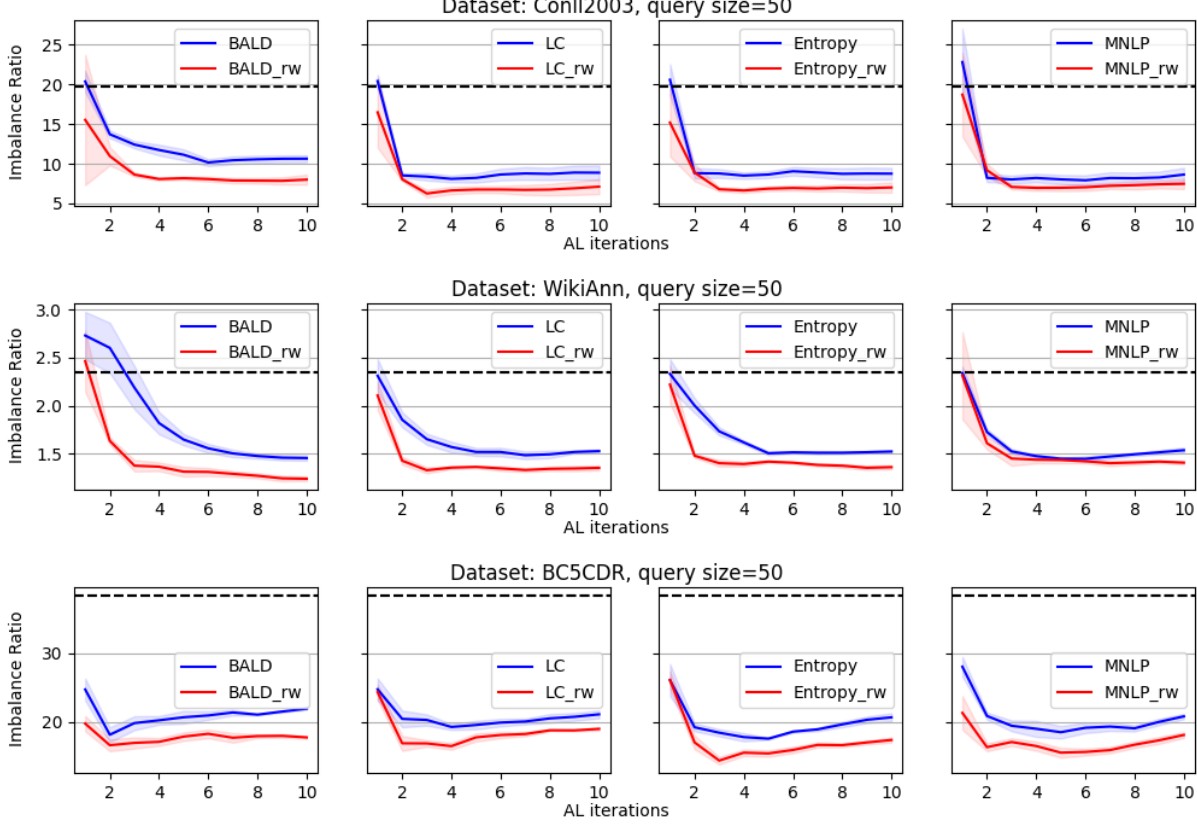

Figure 7: The variation of the imbalance ratio during the active learning process on three datasets with query size 50