# OpenReview forum: "Re-weighting Tokens: A Simple and Effective Active Learning Strategy for Named Entity Recognition"
_EMNLP/2023/Conference — EMNLP 2023 Findings_

### Official Review · Reviewer_xutz · 2023-08-01

**Typos Grammar Style And Presentation Improvements:** The paper is well written.
**Soundness:** 3

**Excitement:**

3: Ambivalent: It has merits (e.g., it reports state-of-the-art results, the idea is nice), but there are key weaknesses (e.g., it describes incremental work), and it can significantly benefit from another round of revision. However, I won't object to accepting it if my co-reviewers champion it.

**Paper Topic And Main Contributions:**

This paper presents a novel active learning strategy that assigns dynamic smoothed weights to individual tokens, contributing to the development of robust active learners for named entity recognition. The paper addresses the challenge of data imbalance in NER and proposes a re-weighting-based active learning strategy that improves the performance of active sequence tagging. The experimental results demonstrate substantial performance improvement, validating its practical efficacy.

**Reasons To Accept:**

This paper proposes a novel active learning strategy for named entity recognition that addresses the challenge of data imbalance and demonstrates its effectiveness through experimental results.

**Reasons To Reject:**

- The paper does not compare the proposed approach with other state-of-the-art active learning strategies for NER, which could provide a more thorough evaluation of its effectiveness.
- The authors do not provide a detailed discussion of the limitations or potential drawbacks of the proposed approach.

**Reproducibility:**

5: Could easily reproduce the results.

**Reviewer Confidence:**

2: Willing to defend my evaluation, but it is fairly likely that I missed some details, didn't understand some central points, or can't be sure about the novelty of the work.

---

> ### Author Rebuttal · Authors · 2023-08-29
>
> We thank reviewer-xutz for the insightful comments and suggestions. We provide a detailed response below.
>
> **Question 1: The paper does not compare the proposed approach with other state-of-the-art active learning strategies for NER, which could provide a more thorough evaluation of its effectiveness.**
>
> Answer 1: We would like to clarify that our study was designed to address sentence-level imbalances and has been compared with leading sentence-level active learning strategies for NER like LC, ENT, MNLP, and BALD. We mentioned other recent state-of-the-art active learning strategies for NER in the first section of the appendix, such as Lowest Token Probability (LTP) [1] and sub-sequence based methods [2] [3]. However, LTP requires the decoder to be conditional random fields, which limits its applicability; while [2] [3] query only a portion of the sentence. These facts make direct comparisons with them infeasible. Studying how to incorporate reweighting mechanisms into these algorithms could be a future direction.
>
> [1] Liu, M., Tu, Z., Wang, Z., & Xu, X. (2020). LTP: a new active learning strategy for BERT-CRF based named entity recognition. arXiv preprint arXiv:2001.02524.
>
> [2] Radmard, P., Fathullah, Y., and Lipani, A. Subsequence based deep active learning for named entity recognition. In Proceedings of the 59th Annual Meeting of the Association for Computational Linguistics and the 11th International Joint Conference on Natural Language Processing (Volume 1: Long Papers), pp. 4310–4321, Online, August 2021. Association for Computational Linguistics. doi: 10.18653/v1/2021.acl-long.332
>
> [3] Liu, Y., Hu, J., Chen, Z., Wan, X., & Chang, T. H. (2023, June). EASAL: Entity-Aware Subsequence-Based Active Learning for Named Entity Recognition. In Proceedings of the AAAI Conference on Artificial Intelligence (Vol. 37, No. 7, pp. 8897-8905).
>
> **Question 2: The authors do not provide a detailed discussion of the limitations or potential drawbacks of the proposed approach.**
>
> Answer 2: We enumerate some limitations of our research here and will include them in the final version of the paper:
> - One of the limitations of our paper is related to the hyperparameter Beta in our reweighting algorithm. We agree that the static nature of Beta could be a drawback, particularly when applying the algorithm to new or different corpora. Furthermore, Beta may also be sensitive to the number of classes in the dataset.
> - We acknowledge that an automated approach to determine the optimal value for Beta would likely yield better results, especially when extending the application of our model to diverse corpora with varying class distributions. We also believe that dynamically updating Beta in line with Active Learning iterations could be a promising avenue for future research.

---

### Official Review · Reviewer_Mdb7 · 2023-08-03

**Soundness:** 3

**Excitement:**

3: Ambivalent: It has merits (e.g., it reports state-of-the-art results, the idea is nice), but there are key weaknesses (e.g., it describes incremental work), and it can significantly benefit from another round of revision. However, I won't object to accepting it if my co-reviewers champion it.

**Paper Topic And Main Contributions:**

Data imbalance is a very big challenge in NER applications.  This paper presents a reweighting-based active learning strategy for active sequence tagging to tackle the data imbalance issue. The proposed method assigns dynamic smoothed weights to individual tokens. This adaptable strategy is compatible with various token-level acquisition functions.  Experimental results on several datasets demonstrate the performance enhancement on NER task.



**Questions For The Authors:**

a.	The hyper-parameter Beta is very important in the re-weighting method. Experimental results on the hyper-parameter beta also should be provided. It would be better to provide an automatic method to optimize the hyper-parameter Beta in the applications.

b.	More details on the experiments should be provided, such as the distribution of each NER type, the precision, recall and F of each type in the active learning process.

c.	Table 1 only provides F1-scores in the first three AL iterations on Conll2003 with query size 15.  How about the performance in the other iterations?


**Reasons To Accept:**

Data imbalance is a very big challenge in NER applications.

This paper presents a reweighting-based active learning strategy for active sequence tagging to tackle the data imbalance issue. The proposed method assigns dynamic smoothed weights to individual tokens. This adaptable strategy is compatible with various token-level acquisition functions.

Experimental results on several datasets demonstrate the performance enhancement on NER task.

**Reasons To Reject:**

The hyper-parameter Beta is very important in the re-weighting method. Experimental results on the hyper-parameter beta also should be provided. It would be better to automatically optimize the hyper0parameter Beta in the applications.

More details on the experiments should be provided, such as the distribution of each NER type, the precision, recall and F of each type in the active learning process.

Table 1 only provides F1-scores  in the first three AL iterations on Conll2003 with query size 15.  How about F1 in the other iterations (i.e. 4-10 iteration)?


**Reproducibility:**

3: Could reproduce the results with some difficulty. The settings of parameters are underspecified or subjectively determined; the training/evaluation data are not widely available.

**Reviewer Confidence:**

5: Positive that my evaluation is correct. I read the paper very carefully and I am very familiar with related work.

---

> ### Author Rebuttal · Authors · 2023-08-29
>
> We thank reviewer-Mdb7 for your constructive suggestions. We provide a detailed response below.
>
> **Question 1: The hyper-parameter Beta is very important in the re-weighting method. Experimental results on the hyper-parameter beta also should be provided. It would be better to provide an automatic method to optimize the hyper-parameter Beta in the applications.**
>
> We acknowledge the absence of study on hyperparameters as a limitation on our part, and we intend to incorporate this aspect in the final version. We suggest, for each dataset, tuning the hyperparameter with a specific base acquisition function and fixing it for other acquisition functions, using the final F1-score on the validation set as a metric. Because the optimal $\beta$ is determined by the imbalanced nature of the current dataset, it can be shared across different acquisition functions. We report the final F1-score for three independent runs after 10 active learning iterations under various hyperparameter settings of $\beta$ on different datasets with batch size=15. Based on the experiments, our method is not particularly sensitive to the choice of hyperparameters. In addition, we found $\beta=0.1$ to be a solid default value, it can work even without excessive hyperparameter tuning.
>
> We acknowledge that an automated approach to determine the optimal value for Beta would likely yield better results, especially when extending the application of our model to diverse corpora with varying class distributions. We also believe that dynamically updating Beta in line with Active Learning iterations could be a promising avenue for future research.
>
> Conll2003:
> | Beta  | 0.01  | 0.1   | 0.2   | 0.5   | 1     |
> |-------|-------|-------|-------|-------|-------|
> | LC_rw | 84.28 | **84.54** | 84.45 | 84.33 | 83.37 |
>
> BC5CDR:
> | Beta  | 0.01  | 0.1   | 0.2   | 0.5   | 1     |
> |-------|-------|-------|-------|-------|-------|
> | LC_rw | 75.50 | **75.98** | 75.90 | 75.63 | 75.51 |
>
> Ontonotes5.0:
> | Beta  | 0.01  | 0.1   | 0.2   | 0.5   | 1     |
> |-------|-------|-------|-------|-------|-------|
> | LC_rw | 76.96 | 77.92 | **78.50** | 78.19 | 77.66 |
>
> **Question 2:  More details on the experiments should be provided, such as the distribution of each NER type, the precision, recall and F of each type in the active learning process.**
>
> We report class frequencies for each NER type and the final F1-score for three independent runs for each class on Conll2003 with a query size of 15, using least confidence as our base acquisition function. It can be observed that the re-weighting version can significantly improve the performance of the minority classes without overly affecting the performance of the majority classes. This can explain why it improves the overall performance. We will include these details on the experiments in the final version.
>
> |            | O      | B-LOC  | I-LOC  | B-PER  | I-PER  | B-MISC | I-MISC | B-ORG  | I-ORG  |
> |------------|--------|--------|--------|--------|--------|---------|--------|--------|--------|
> | Class proportion | 0.833  | 0.035  | 0.0057 | 0.0324 | 0.0222 | 0.0169  | 0.0057 | 0.0310 | 0.0182 |
> | LC_rw      | **98.84** | **82.30** | **70.94** | 92.77 | 95.95 | 69.97  | **63.51** | **80.23** | **80.65** |
> | LC         | 98.67 | 79.66 | 64.82 | **93.56** | **97.16** | **71.39**  | 59.89 | 76.86 | 79.24 |
>
> **Question 3: Table 1 only provides F1-scores in the first three AL iterations on Conll2003 with query size 15. How about the performance in the other iterations?**
>
> We report a comparison of the final F1-score for three independent runs between the smoothed and unsmoothed versions on Conll2003 with batch size=15, using least confidence as our base acquisition function. It can be observed that although the final performances are close, the unsmoothed version exhibited poorer performance and higher fluctuations during the early iterations of active learning, aligning with the purpose of introducing a smoothing function to counteract the high variance in the early stages of active learning.
>
> | iteration | 1     | 2     | 3     | 4     | 5     | 6     | 7     | 8     | 9     | 10    |
> |-----------|-------|-------|-------|-------|-------|-------|-------|-------|-------|-------|
> | +smooth   | 67.43 | **77.02** | **79.93** | 80.86 | **81.64** | **82.64** | **81.56** | **83.22** | **82.47** | **83.46** |
> | no smooth   | **68.73** | 73.96 | 76.83 | **81.15** | 80.46 | 80.07 | 81.32 | 82.45 | 82.37 | 82.97 |

---

### Official Review · Reviewer_9oCT · 2023-08-04

**Soundness:** 3

**Excitement:**

4: Strong: This paper deepens the understanding of some phenomenon or lowers the barriers to an existing research direction.

**Paper Topic And Main Contributions:**

This paper proposes an active learning strategy, "reweighting tokens," suitable for Named Entity Recognition (NER). The strategy is simple yet widely applicable. The authors conducted experiments to demonstrate how this approach effectively addresses class imbalance issues and improves NER model performance.

**Questions For The Authors:**

The experimentation lacks diversity in the number of classes for the datasets. Introducing results for Twitter or Ontonote datasets would strengthen the argument.

More studies on the hyperparameter "beta" in the experimental section would provide better guidance in selecting an appropriate beta value.

**Reasons To Accept:**

The article is well-written, logically structured, and easily understandable.

The method is straightforward and has broad applicability.

**Reasons To Reject:**

The choice of hyperparameter "beta" is very important. The experiment lacks studies and guidelines for how to choose beta properly.

**Reproducibility:**

5: Could easily reproduce the results.

**Reviewer Confidence:**

5: Positive that my evaluation is correct. I read the paper very carefully and I am very familiar with related work.

---

> ### Author Rebuttal · Authors · 2023-08-29
>
> We thank reviewer-9oCT for the constructive feedback. We provide a detailed response below.
>
> **Question 1: The experimentation lacks diversity in the number of classes for the datasets. Introducing results for Twitter or Ontonote datasets would strengthen the argument.**
>
> To address this concern, we conducted additional experiments on Ontonotes5.0 with a query size of 15 and reported the average final F1-score for three independent runs after 10 active learning iterations. It can be observed that the re-weighting method can consistently improve the performance over different baselines. In the revised version of the paper, we are open to including the learning curves for different batch sizes on Ontonotes5.0, both in the main text and in the appendix, for a more comprehensive understanding.
>
> |         | Bald  | LC    | Entropy | MNLP  | Random |
> |---------|-------|-------|---------|-------|--------|
> | With rw | **76.34** | **76.73** | **77.38**   | **77.50** | 59.42  |
> | w/o rw | 73.65 | 76.44 | 76.45   | 76.76 |        |
>
> **Question 2: More studies on the hyperparameter "beta" in the experimental section would provide better guidance in selecting an appropriate beta value.**
>
> We acknowledge the absence of study on hyperparameters as a limitation on our part, and we intend to incorporate this aspect in the final version. We suggest, for each dataset, tuning the hyperparameter with a specific base acquisition function and fixing it for other acquisition functions, using the final F1-score on the validation set as a metric. Because the optimal $\beta$ is determined by the imbalanced nature of the current dataset, it can be shared across different acquisition functions. We report the final F1-score for three independent runs after 10 active learning iterations under various hyperparameter settings of $\beta$ on different datasets with batch size=15. Based on the experiments, our method is not particularly sensitive to the choice of hyperparameters. In addition, we found $\beta=0.1$ to be a solid default value, it can work even without excessive hyperparameter tuning.
>
> Conll2003:
> | Beta  | 0.01  | 0.1   | 0.2   | 0.5   | 1     |
> |-------|-------|-------|-------|-------|-------|
> | LC_rw | 84.28 | **84.54** | 84.45 | 84.33 | 83.37 |
>
> BC5CDR:
> | Beta  | 0.01  | 0.1   | 0.2   | 0.5   | 1     |
> |-------|-------|-------|-------|-------|-------|
> | LC_rw | 75.50 | **75.98** | 75.90 | 75.63 | 75.51 |
>
> Ontonotes5.0:
> | Beta  | 0.01  | 0.1   | 0.2   | 0.5   | 1     |
> |-------|-------|-------|-------|-------|-------|
> | LC_rw | 76.96 | 77.92 | **78.50** | 78.19 | 77.66 |

---

### Meta-Review · Area_Chair_FVba · 2023-09-19

**Recommendation:** 3

**Metareview:**

The paper proposes a reweighting-based active learning strategy for sequence tagging, with a focus on named entity recognition (NER) tasks. While the paper exhibits strengths, there are important aspects that need attention and further clarification. Below I summarize the most relevant aspects to be considered:
Reasons to Accept:
* 		Clarity and Readability: The paper is well-written, logically structured, and easily understandable. The presentation of the proposed method is clear, contributing to the overall comprehensibility of the research.
* 		Broad Applicability: The method presented in the paper is straightforward and exhibits potential for broad applicability. Its compatibility with various token-level acquisition functions is a valuable feature, demonstrating versatility.
* 		Innovative Active Learning Strategy: The paper introduces a novel active learning strategy for NER that effectively addresses the challenge of data imbalance. The proposed dynamic smoothed weights for tokens represent an adaptable approach that holds promise for improving sequence tagging tasks.
* 		Empirical Validation: The paper substantiates its claims through experimental results on multiple datasets, showcasing performance enhancements in the context of NER tasks. This empirical validation enhances the credibility of the proposed approach.
Reasons to Reject:
* 		Limited Hyperparameter Analysis: The absence of a thorough study on hyperparameters, specifically the importance of Beta in the re-weighting method, is a notable limitation. It is essential to explore the sensitivity of the method to hyperparameter variations and provide insights into the optimal choices for different scenarios.
* 		Lack of Hyperparameter Optimization: The paper could benefit from experimental results and discussions related to the hyperparameter Beta. Automatic optimization or at least a systematic exploration of this hyperparameter's impact on performance would strengthen the method's practical utility.
* 		Static Nature of Beta: The authors acknowledge the static nature of the Beta hyperparameter, which may pose limitations when applying the algorithm to new or diverse corpora. Further investigation into the adaptability of Beta and its sensitivity to dataset characteristics, such as the number of classes, is essential.
* 		Additional Experiment Details: While the paper presents experimental results demonstrating overall performance enhancement, it could be further strengthened by providing more detailed metrics, including the distribution of each NER type and precision, recall, and F1 scores for each type throughout the active learning process. Such details would offer a deeper understanding of the method's effectiveness.
In conclusion, while the paper introduces an active learning strategy for NER with several strengths, including clarity, applicability, and empirical validation, there are areas that require attention. These areas were discussed during the rebuttal period and there was convergence on the opinions. Thus, more specifically, addressing hyperparameter sensitivity, exploring automatic hyperparameter optimization, and providing additional experiment details would enhance the paper's overall quality and practicality.

---

### Decision · Program_Chairs · 2023-10-07

**Decision:**

Accept-Findings

**Comment:**

The paper proposes a reweighting-based active learning strategy for sequence tagging, with a focus on named entity recognition (NER) tasks. While the paper exhibits strengths, there are important aspects that need attention and further clarification. Below I summarize the most relevant aspects to be considered:
Reasons to Accept:
* 		Clarity and Readability: The paper is well-written, logically structured, and easily understandable. The presentation of the proposed method is clear, contributing to the overall comprehensibility of the research.
* 		Broad Applicability: The method presented in the paper is straightforward and exhibits potential for broad applicability. Its compatibility with various token-level acquisition functions is a valuable feature, demonstrating versatility.
* 		Innovative Active Learning Strategy: The paper introduces a novel active learning strategy for NER that effectively addresses the challenge of data imbalance. The proposed dynamic smoothed weights for tokens represent an adaptable approach that holds promise for improving sequence tagging tasks.
* 		Empirical Validation: The paper substantiates its claims through experimental results on multiple datasets, showcasing performance enhancements in the context of NER tasks. This empirical validation enhances the credibility of the proposed approach.
Reasons to Reject:
* 		Limited Hyperparameter Analysis: The absence of a thorough study on hyperparameters, specifically the importance of Beta in the re-weighting method, is a notable limitation. It is essential to explore the sensitivity of the method to hyperparameter variations and provide insights into the optimal choices for different scenarios.
* 		Lack of Hyperparameter Optimization: The paper could benefit from experimental results and discussions related to the hyperparameter Beta. Automatic optimization or at least a systematic exploration of this hyperparameter's impact on performance would strengthen the method's practical utility.
* 		Static Nature of Beta: The authors acknowledge the static nature of the Beta hyperparameter, which may pose limitations when applying the algorithm to new or diverse corpora. Further investigation into the adaptability of Beta and its sensitivity to dataset characteristics, such as the number of classes, is essential.
* 		Additional Experiment Details: While the paper presents experimental results demonstrating overall performance enhancement, it could be further strengthened by providing more detailed metrics, including the distribution of each NER type and precision, recall, and F1 scores for each type throughout the active learning process. Such details would offer a deeper understanding of the method's effectiveness.
In conclusion, while the paper introduces an active learning strategy for NER with several strengths, including clarity, applicability, and empirical validation, there are areas that require attention. These areas were discussed during the rebuttal period and there was convergence on the opinions. Thus, more specifically, addressing hyperparameter sensitivity, exploring automatic hyperparameter optimization, and providing additional experiment details would enhance the paper's overall quality and practicality.